# Preparation and Characterization of Phenolic Acid-Chitosan Derivatives as an Edible Coating for Enhanced Preservation of Saimaiti Apricots

**DOI:** 10.3390/foods11223548

**Published:** 2022-11-08

**Authors:** Fangjie Li, Yingying Yan, Chengzhi Gu, Jiaying Sun, Yaru Han, Zhaoqing Huangfu, Fangyuan Song, Jiluan Chen

**Affiliations:** 1College of Food, Shihezi University, Xinjiang Uygur Autonomous Region, Shihezi 832000, China; 2College of Chemistry and Chemical Engineering, Shihezi University, Xinjiang Uygur Autonomous Region, Shihezi 832000, China

**Keywords:** chitosan, phenolic acids, antioxidant activity, antimicrobial activity, apricot

## Abstract

In this study, caffeic acid (CA) and chlorogenic acid (CGA) were incorporated onto chitosan (CS) using free radical grafting initiated by a hydrogen peroxide/ascorbic acid (H_2_O_2_/Vc) redox system. The structural properties of the CA (CA-g-CS) and CGA (CGA-g-CS) derivatives were characterized by UV–Vis absorption, Fourier transform infrared spectroscopy (FTIR), X-ray diffraction (XRD), nuclear magnetic resonance (NMR), and thermal stability analysis. Then, the antioxidant and antibacterial properties were evaluated, and the effect of CGA-g-CS on the postharvest quality of Saimaiti apricot was studied. It proved that phenolic acids were successfully grafted onto the CS. The grafting ratios of CA-g-CS and CGA-g-CS were 126.21 mg CAE/g and 148.94 mg CGAE/g. The antioxidation and antibacterial activities of CGA-g-CS were better than those of CA-g-CS. The MICs of CGA-g-CS against *E. coli*, *S. aureus*, and *B. subtilis* were 2, 1, and 2 mg/mL. The inhibitory zones of 20 mg/mL CGA-g-CS against the three bacteria were 19.16 ± 0.35, 16.33 ± 0.91, and 16.24 ± 0.05 mm. The inhibitory effects of 0.5% CGA-g-CS on the firmness, weight loss, SSC, TA, relative conductivity, and respiration rate of the apricot were superior. Our results suggest that CGA-g-CS can be potentially used as an edible coating material to preserve apricots.

## 1. Introduction

Apricot (*Armeniaca vulgaris* Lam.) is a deciduous plant belonging to the Rosaceae family. The apricot fruit can be eaten either fresh or processed. Xinjiang is one of the largest apricot-producing areas in China, where abundant, good-quality apricot varieties are grown. Among these varieties, Saimaiti apricots are favored by consumers due to their bright color, tender juicy flesh, sweet and sour taste, unique flavor, and nutrient richness [1]. However, apricots typically exhibit vigorous respiration and other physiological activities after harvesting, which causes a loss of nutrients and flavoring substances. This reduces their nutritional quality, commodity value, and market share, which restricts the marketability of apricot. In addition to storage at low temperatures and other preservation methods, apricots can be preserved using novel edible coatings containing chitosan, proteins, lipids, and other materials, which form a semipermeable protective barrier on the fruits’ surface. This enhances the quality of the fruits and prolongs the storage period [2,3].

Chitosan (CS) is a natural cationic polysaccharide composed of β-(1-4)-2-amino-D-glucose and β-(1-4)-2-acetylamino-D-glucose [4]. CS is increasingly utilized as a promising biomaterial due to its biocompatibility, biodegradability, nontoxicity, and antibacterial and antioxidant properties [5]. It is suitable for use in food, medicine, and cosmetics [6,7,8]. However, its poor aqueous solubility limits its utilization in industries. There are three types of functional groups on the CS backbone, including amino/acetamido, primary, and secondary hydroxyl groups at the C-2, C-3, and C-6 positions, respectively. Their arrangement and distribution influence the structural, physical, and chemical properties of CS [9,10]. Its functional properties can be improved via chemical modifications by introducing amino and hydroxyl groups into the CS molecule under suitable conditions [11,12].

Caffeic acid (CA) and chlorogenic acid (CGA) are naturally occurring phenolic acids extracted from plants. They have ortho hydroxyl groups on the aromatic residues, and have antioxidant, anti-inflammatory, immune regulation, antiallergy, antiatherosclerosis, antithrombosis, cardiovascular protection, anticarcinogenic, and antidiabetic properties [13,14]. Moreover, phenolic acids can destabilize microbial membranes and block protein synthesis in microbial cells, resulting in cell death, thus making them good antibacterial agents [15]. Therefore, the grafting of phenolic acids onto CS chains can potentially enhance their antioxidant and antibacterial properties [16]. Currently, there are four major methods for preparing phenolic acid–CS conjugates, including carbodiimide coupling, enzyme-catalyzed, electrochemical, and free-radical-mediated grafting methods [17,18,19,20]. Carbodiimide coupling reactions usually require several chemical crosslinking reagents, which are used to add CS derivatives to foods. They have potential adverse effects on humans [21,22]. Enzymatic catalysis methods (using laccases and tyrosinases) and electrochemical methods (voltage catalysis) are both used to oxidize phenolic acids to O-quinones, which are then grafted onto CS by Michael addition and Schiff base reaction to synthesize CS/phenolic acid graft copolymers [20,23,24,25]. However, phenolic acids are easily oxidized during the reaction process, which reduces the antioxidant and antibacterial activities of the grafted copolymers. The free-radical-induced grafting method uses a hydrogen peroxide/ascorbic acid (H_2_O_2_/Vc) redox system and is considered an efficient, low-cost, and green method because the reactions occur at room temperature without producing toxic substances. Currently, there are two theories about the possible mechanism of free-radical-mediated grafting. Liu et al. [26] used EPR technology to demonstrate, for the first time, that the graft copolymerization of CS with phenolic acids using the H_2_O_2_/Vc redox system is mediated by ascorbic acid (Asc•−) rather than hydroxyl (•OH), verifying that •OH cannot generate any active sites on the CS chain. This provides further information regarding the mechanism of phenolic acid–CS grafting. Although several studies have used free radical grafting to graft phenolic acid onto CS chains [27,28], it has rarely been used to compare two phenolic acids and their functionality simultaneously.

The aim of this study was to prepare CS derivatives with CA (CA-g-CS) and CGA (CGA-g-CS) and characterize their structures. We also evaluated the antioxidant and antimicrobial properties of these two CS derivatives and investigated their effects on the storage quality of Saimaiti apricots.

## 2. Materials and Methods

### 2.1. Materials and Reagents

CS (average molecular weight = 130 KDa, degree of deacetylation ≥90%) were purchased from Bozhi Huili Biotechnology Co. Ltd. (Qingdao, China). CA, CGA, Folin–Ciocalteau reagent, 2,2′-azino-bis (3-ethylbenzothiazoline-6)-sulfonic acid (ABTS), and 2,2-diphenyl-1-picrylhydrazyl (DPPH) were purchased from Maclin Biotechemical Co. Ltd. (Shanghai, China). The bacterial strains *Escherichia coli* (CMCC 44102), *Staphylococcus aureus* (CMCC 26003), and *Bacillus subtilis* (CMCC 63501) were provided by the Microbiology Laboratory of Shihezi University. All other reagents were of analytical grade.

### 2.2. Preparation of CA-g-CS and CGA-g-CS

CA-g-CS and CGA-g-CS were prepared using a H_2_O_2_/Vc redox pair in an inert atmosphere based on the method described by Curcio [29], with some modifications. Briefly, 0.5 g of CS was dissolved in 50 mL of 1% acetic acid (*v*/*v*) in a 250 mL, three-necked, round-bottomed flask. After stirring continuously for 8 h (500 rpm), 3 mL of 1 M H_2_O_2_ and 0.27 g Vc were added and constantly stirred for 30 min. Then, 0.2 g phenolic acid (CA or CGA) was added and reacted for 24 h. The whole reaction took place under N_2_ protection. After the reaction, the solution was dialyzed using an 8–14 kDa dialysis bag to remove the unreacted phenolic acid, ascorbic acid, and other small molecular compounds. The CS derivatives were prepared by freeze-drying after dialysis for 72 h (the dialysis water was changed every 12 h).

### 2.3. Determination of the Grafting Ratios

The grafting ratios of CA-g-CS and CGA-g-CS were measured using the Folin–Ciocalteu method, with slight modifications [30]. Briefly, 40 μL of CS derivative (1 mg/mL) was mixed with 200 μL of Folin–Ciocalteu reagent and 1160 μL distilled water and allowed to react at 30 °C for 5 min in the dark. Then, 600 μL of 20% Na_2_CO_3_ solution was added, and the mixture was allowed to stand for 2 h before reading the absorbance at 760 nm. CA and CGA were used to calculate the standard curves. The grafting ratios of CA-g-CS and CGA-g-CS were expressed as mg of the CA equivalents per g (mg CAE/g) and mg of the CGA equivalents per g (mg CGAE/g), respectively.

### 2.4. Structural Characterization of CA-g-CS and CGA-g-CS

To ensure that CA and CGA were successfully grafted onto CS, the structures of CA-g-CS and CGA-g-CS were characterized using UV–Vis spectrum and Fourier transform infrared (FT-IR) spectroscopy and nuclear magnetic resonance (^1^H NMR and ^13^C NMR) spectroscopy analysis [31]. The UV–Vis spectra of CS, CA-g-CS, and CGA-g-CS were recorded using a UV–Vis spectrophotometer (UV-2600, Shimadzu, Kyoto, Japan) in the range of 210–500 nm. The FT-IR spectra were analyzed using a continuous-scan FT-IR spectrometer (Nicolet IR200, Thermo, USA) in the frequency range of 4000–400 cm^−1^ using the KBr-disks method. The 1H NMR analysis was conducted with samples dissolved in CF_3_COOD/D_2_O solution (1%, *v*/*v*) using a 400 MHz NMR spectrometer (Ascend 400, Bruker, Switzerland).

The crystal behavior of the phenolic acid-g-CSS was determined by X-ray diffraction (XRD) with a Bruker AXS D8 Advance X-ray diffractometer at 2θ = 10–80° (Bruker Inc., Bremen, Germany). The thermogravimetric (TG) analysis of CA-g-CS and CGA-g-CS was performed using TGA (NETZSCH STA 449F3, Selb, Germany) and by heating the samples from 25 °C to 800 °C in 10 °C/min increments under nitrogen protection.

### 2.5. Antioxidant Activity Assay

The antioxidant activities of the conjugates were evaluated using a series of concentrations of 0.25, 0.5, 1, 2, and 4 mg/mL using several methods, as follows. The DPPH radical scavenging activity was evaluated according to the method of Xie et al. [32]. The ABTS scavenging activity was measured using the method of Kang et al. [33]. The hydroxyl radical scavenging activity and reducing power were evaluated according to the methods of Zhong et al. [34] and Oyaizu et al. [35], respectively.

### 2.6. Antimicrobial Activity Assay

We evaluated the antimicrobial properties of the phenolic acid-g-CSs using the minimum inhibitory concentration (MIC) and inhibition zone methods. *E. coli*, *S. aureus*, and *B. subtilis* were selected, since they are commonly found in fruits and vegetables. We inoculated the strain in LB solid medium with an inoculation ring, put it into a 37 °C incubator for 12 h after scribing, and repeated this step until a single colony appeared. We inoculated the single colony in 100 mL LB liquid medium and incubated it on a shaking table at 37 °C at 160 r/min for 12 h. Then, we took 1 mL of inoculum and put it into 100 mL of LB liquid culture medium. We incubated it on a 160 r/min shaking table at 37 °C for 10 h. The bacterial growth was determined at OD_600_ nm, and the number of colonies was around 10^6–^10^7^ CFU/mL. Then, we put it into a 4 °C refrigerator for temporary storage.

The MIC was determined according to the methods of Wang et al. [36]. Briefly, we prepared sample solutions with concentrations of 0.125, 0.25, 0.5, 1.0, 2.0, and 4.0 mg/mL, and then mixed 15 mL of nutrient agar before its solidification with 1 mL of the sample solution to cause the mixture to solidify at room temperature. The bacteria culture (100 μL) grown in a Petri dish was incubated at 37 °C for 24 h.

The inhibition zone was measured according to the methods of Su et al. [37]. We took 200 μL of bacterial solution and evenly spread it on the surface of the solid medium. Then, we placed three oxford cups in the culture dishes. We injected 200 μL of the sample solution (10, 15, and 20 mg/mL) into each oxford cup, and then put the culture dishes in the 37 °C incubator for 24 h. The inhibition circle was measured with a vernier caliper.

### 2.7. In Vivo Assay

By comparing the structural characterization, antioxidation, and antibacterial activities of the phenolic acid-chitosan, the better performance was selected, and the fresh-keeping effect of this phenolic acid-chitosan on the Saimaiti apricot was further evaluated.

#### 2.7.1. Preparation of the Chitosan Solution and Treatment of the Apricots

The apricots (*Armeniaca vulgaris* Lam. Saimaiti) were harvested from an orchard in Kashi (Xinjiang, China). We selected fruits that were uniform in shape and size, had no mechanical damage, and were of a similar ripeness. These fruits were randomly divided into 5 groups (*n* = 200/group), and three replicates were used per group. To prepare the chitosan and phenolic acid-chitosan derivatives, solutions with concentrations of 0.1% and 0.5% (*w*/*v*) and 2.0 g and 10.0 g of chitosan and phenolic acid-chitosan derivative were, respectively, dissolved in 1.8 L 1.0% acetic acid solution, and then we adjusted the pH of the chitosan solution to 5.6 with 2.0 mol/L NaOH solution. The solution was supplemented to 2.0 L, and 0.1% Tween 80 was added as the surfactant. The fruits were soaked in distilled water (CK) with 0.1% or 0.5% CS or phenolic acid-g-CS solution for 5 min and then dried in air. A coating was formed on the apricots treated with chitosan and the phenolic acid-chitosan derivatives. Then, they were placed in plastic baskets and stored at 1 °C in 80–90% relative humidity for 35 days. During the storage period, 40 fruits were taken out every 7 days in triplicate to determine the weight loss rate, firmness, soluble solid solution content (SSC), titratable acidity (TA), relative electrical conductivity, and respiration rate. Each experiment was performed at least twice.

#### 2.7.2. Determination of the Weight Loss Rate

The weight loss rate of the apricots was determined by the weighing method. In each treatment group, we marked out 200 apricots for weighing and then weighed them every 7 days. The calculation formula is as follows:$$\mathrm{Weight}\;\mathrm{loss}\;\mathrm{rate}\left(\%\right)=\frac{\mathrm{Initial}\;\mathrm{weight}\;\mathrm{of}\;\mathrm{fruit}-\mathrm{weight}\;\mathrm{of}\;\mathrm{fruit}\;\mathrm{at}\;\mathrm{each}\;\mathrm{sampling}\;}{\mathrm{Initial}\;\mathrm{weight}\;\mathrm{of}\;\mathrm{fruit}}$$

#### 2.7.3. Determination of the Firmness

A total of 20 fruits in each group were randomly selected and tested with a GY-4 fruit sclerometer with a probe diameter of 2 mm. The firmness values were measured around and along the equator of the apricots, and the unit was N.

#### 2.7.4. Determination of Soluble Solid Solution Content (SSC) and Titratable Acidity (TA)

The soluble solid solution contents (SSC) were determined using a hand-held digital refractometer. The titratable acids (TA) were analyzed according to the method of Batista Silva et al. [38]. Five fruits were randomly selected from each group for the analysis.

#### 2.7.5. Determination of the Relative Electrical Conductivity

The samples were cut into 2 mm slices and oscillated in ultrapure water for 20 min, whereafter they were rinsed with ultrapure water. We measured the conductivity value 20 min before and after boiling, and the specific value was the relative conductivity.

#### 2.7.6. Determination of the Respiration Rate

The respiration rate was expressed as the mass of CO_2_ released per kilogram per hour by the apricots [39].

### 2.8. Statistical Analysis

SPSS 19.0 (SPSS Inc., Chicago, IL, USA) software was used for the statistical analysis. Single-factor analysis of variance (ANOVA) and Duncan’s multiple range tests (*p* < 0.05) were used to determine the significant differences between samples. Origin 8.1 (Microcal Software Inc., Northampton, MA, USA) software was used to for the drawing.

## 3. Results and Discussion

### 3.1. Synthesis of Phenolic Acid-g-CSs

In this study, CA and CGA were successfully grafted onto the CS chains using the Vc and H_2_O_2_ redox pair under N_2_. As reported previously [26], Asc• can adopt hydrogen atoms and form carbon radicals on the CS chains, significantly weakening the intramolecular and intermolecular hydrogen bonding. Subsequently, a large number of CA or CGA monomers can be grafted onto CS, which further weakens the hydrogen bonding in CS. The grafting ratios of CA-g-CS and CGA-g-CS were determined as 126.21 mg CAE/g and 148.94 mg CGAE/g, respectively, using the Folin–Ciocalteu method. The grafting ratio obtained in this study was higher than that of other phenolic acids grafted using the same method [30,40], possibly due to the higher H_2_O_2_/Vc ratio used.

### 3.2. UV–Vis Spectra of Phenolic Acid-g-CSs

Figure 1A shows the UV–Vis spectra of CS, CA-g-CS, and CGA-g-CS. CS did not display any absorption peaks between 210 and 500 nm. However, CA-g-CS showed bands at 295 and 328 nm, while CGA-g-CS showed bands at 296 and 322 nm, respectively. This indicated that CA and CGA were successfully grafted onto CS. Compared with the characteristic peaks of CA at 293 and 320 nm, the UV–Vis absorption peaks of CA-g-CS exhibited a red shift, which might be due to the covalent connection between CA and CS that reduces the energy required for n-π* and π-π* transitions. CGA and CGA-g-CS have characteristic peaks at 322 nm, indicating a lack of red shift phenomena. This is consistent with the results obtained by Zhang [41] and might be due to the distance between the aromatic ring of CGA and the graft point. However, it does not enhance the conjugated system [17].

### 3.3. Fourier Transform Infrared (FT-IR) of Phenolic Acid-g-CSs

Figure 1B shows the FT-IR spectra of CS, CA-g-CS, and CGA-g-CS. The intense band of CS in the range of 3200–3500 cm^−1^ is caused by the N–H and O–H stretching vibrations of the amino acid-polysaccharide molecules [17]. The bands at 1650, 1550, and 1320 cm^−1^ correspond to the N-acetyl residues, including C=O stretching (amide I), N–H bending (amide II), and C–N stretching (amide III), respectively. The band at 1595 cm^−1^ was attributed to the N–H bending of the primary amine [40]. Compared with CS, the grafted polymers exhibited the characteristic band for the aromatic ring C=C in the phenolic acid at 1450–1600 cm^−1^. However, no bands were observed at 1595 cm^−1^, indicating that the NH_2_ on the CS chain underwent a covalent coupling reaction, which converted the primary amine into a secondary amine [35]. Additionally, the bands at 1320 and 1420 cm^−1^ indicated that the –OH of the phenolic acids and the –NH_2_ of CS form amide bonds. However, CA-g-CS exhibited a band at 1735 cm^−1^, indicating the formation of an ester bond between the –OH of CS and –COOH of CA. These results suggest that the conjugation reaction mainly occurred at the –COOH of CA, –OH of C-6, and –NH_2_ of C-2 on CS, while CGA reacts with the C-2 position on CS.

### 3.4. NMR Spectra of the Phenolic Acid-g-CSs

The ^1^H NMR spectra of CS, CA-g-CS, and CGA-g-CS show a difference between the NMR spectra of CS and its derivatives (Figure 2A). The single peaks for CS at 2.9, 4.6, and 2 ppm represent the hydrogen atoms of the C-2, C-1, and acetyl N–H groups, respectively. Multiple peaks between 3.6 and 3.9 ppm represent hydrogen atoms between C-3 and C-6 [42]. In addition to the characteristic peak of CS, several new peaks appeared, which corresponded to the hydrogen atoms on the benzene ring of CGA at 7 ppm, C-e hydrogen atoms at 6.1 ppm, C-d hydrogen atoms at 7.3 ppm, and C-g/h hydrogen atoms at 2.1 ppm [43]. Similarly, five characteristic peaks were observed between 6 and 7.6 ppm on the CA-g-CS map. These represent five hydrogen atoms on the CA benzene ring and carbon–carbon double bond, which were at 7.54 ppm on C-g, 7.2 ppm on C-b, 7.05 ppm on C-f, 6.87 ppm on C-e, and 6.3 ppm on C-h [44]. These experimental results further confirmed the successful grafting of CGA and CA onto CS.

CS and its polymers were further characterized by liquid-state ^13^C NMR. As shown in Figure 2B, the 55.5, 59.7, 76.1, 74.5, 69.6, and 97.3 ppm signals corresponded to the C-2, C-6, C-3, and C-5, C-4, and C-1 carbons of CS, respectively. Additionally, the carbonyl and methyl groups of N-acetylglucosamine (C-7 and C-8) were at 176.4 ppm and 20.1 ppm, respectively [42]. The C-1, C-3–C-5, and C-6 peaks of CA-g-CS and CGA-g-CS were larger than those of CS. The new peaks at 144, 122.4, and 116 ppm correspond to the C=C in the derivative. The increased signal intensity at 171 and 174.5 ppm was due to the carbonyl group (C=O) formation between the amino group of CS and the carboxyl group of phenolic acid. The results further showed that CA and CGA were successfully grafted onto CS.

### 3.5. XRD of Phenolic Acid-g-CSs

The crystallographic structures of CS, CA-g-CS, and CGA-g-CS were determined by XRD. Various intramolecular and intermolecular hydrogen bond interactions formed between the –OH, H–NH_2_, and –NH–COCH_3_ functional groups in the CS chain inhibit the rotation of the adjacent sugar residues along the glycosidic bond, resulting in the formation of crystals [45]. Figure 3A shows the two crystal forms of CS, known as form I (2θ = 11.5°) and form II (2θ = 20°), which formed due to the hydrogen bonds in the CS chains [46]. However, the diffraction peaks of CA-g-CS and CGA-g-CS were broader and weaker at 2θ = 21.9° and 20.5°, respectively. The decrease in the CS crystallinity might be due to the introduction of the CA and CGA residues, significantly reducing the force of the hydrogen bonds on the CS chain [35]. This also proved that phenolic acid was successfully grafted onto the CS, consistent with previous reports [22,47].

### 3.6. Thermal Properties of Phenolic Acid-g-CSs

The thermal properties of CS, CA-g-CS, and CGA-g-CS were investigated using TGA to quantitatively measure the mass change in the samples with the increase in temperature. Figure 3B shows that the TGA curve of CS exhibited two weight loss stages. The first was observed in the range of 25–155 °C, possibly due to the loss of water from the polymer composites. The second was between 270 °C and 800 °C and was due to the pyrolytic decomposition of CS [48]. CA-g-CS and CGA-g-CS also exhibited these two stages. The first stage occurred from 25 to 148 °C and 25 to 141 °C for CA-g-CS and CGA-g-CS, respectively, due to the loss of the adsorbed and bound water. The second stage occurred between 149 and 800 °C and 142 and 800 °C for CA-g-CS and CGA-g-CS, respectively, due to the degradation of the grafted products. Pasanphan and Chirachanchai proposed that the grafting of phenolic acids to CS hinders the original arrangement of the CS chain, which decreases its thermal stability [49].

### 3.7. Antioxidant Activity of Phenolic Acid-g-CSs

The antioxidant activity of CS and its derivatives is mainly due to the inherent free radical scavenging activity of the phenolic acids. We used DPPH, ABTS, and hydroxyl radical scavenging power and reducing power to characterize the antioxidant capacity. Antioxidants convert stable DPPH free radical (purple) to the non-free radical form DPPH-H (yellow) [50]. The scavenging capacity of DPPH radical is closely related to the hydrogen supply capacity of antioxidants [32,51]. Figure 4A shows that the oxidation resistance of the derivatives was significantly (*p* < 0.05) higher than that of CS, possibly due to the chemically modified phenolic acid groups on CS, which improved its hydrogen or electron supply capacity [44]. The EC_50_ values of CS, CA-g-CS, and CGA-g-CS were 125.89, 52.45, and 41.96 mg/mL, respectively, indicating that the DPPH scavenging activity of CA-g-CS and CGA-g-CS was 2.4-fold and 3.0-fold higher than that of CS. The antioxidant capability was positively correlated with the concentration. At a 4 mg/mL concentration, the scavenging rate of CA-g-CS and CGA-g-CS reached 75.64% ± 0.24% and 84.27% ± 0.44%, respectively, compared to 8.47% ± 1.37% for CS. Moreover, the DPPH radical scavenging activity of the phenolic acid-CSs showed that the higher the grafting rate was, the higher the scavenging activity was, consistent with the results shown by Aytekin et al. [21].

The ABTS free radical method is widely used to determine the antioxidant activity, which is evaluated based on the intensity of the color change in the ABTS radicals during the reaction [17]. The polymers showed a concentration-dependent increase in their antioxidant activity (Figure 4B). The EC_50_ value of CS was 50.12 mg/mL. After introducing CA and CGA, the EC_50_ values of CA-g-CS and CGA-g-CS were 20.89 and 14.82 mg/mL, respectively, indicating that the antioxidant activity of the phenolic acid-chitosan derivatives was higher than that of CS. CA-g-CS had a higher ABTS radical scavenging ability than CGA-g-CS (*p* < 0.05). At 4 mg/mL, the scavenging rates of CS, CA-g-CS, and CGA-g-CS were 22.32% ± 2.78%, 65.64% ± 2.37%, and 77.8% ± 0.2%, respectively. According to the XRD analysis, the grafting reaction led to the loosening of the CS chain and the increase in the water solubility of CS, thus exposing more active sites related to the H atom donors. This may be the reason for the increased free radical scavenging capacity of the phenolic acid-chitosan derivatives.

Hydroxyl radicals can extensively oxidize amino acids, proteins, DNA, and other macromolecules, leading to cell death. Figure 4C shows that the two derivatives had a stronger scavenging activity than CS (*p* < 0.05). The EC_50_ value of CS was 79.43 mg/mL, while those of CA-g-CS and CGA-g-CS were 33.06 and 25.75 mg/mL, respectively. The hydroxyl radical scavenging activity was 2.4-fold and 3.1-fold greater than that of CS. At 4 mg/mL, the scavenging rates of CS, CA-g-CS, and CGA-g-CS were 43.85% ± 0.14%, 79.82% ± 1.54%, and 85.50% ± 0.69%, respectively. Similar results were observed in gallic-acid-grafted chitosan [32].

Evaluating the reducing power is important for characterizing the electron-donating capacity of antioxidants. This is evaluated based on the antioxidants’ capacity to inhibit the transformation of divalent iron to trivalent iron by providing electrons, which can react with free radicals. The absorbance of the samples was measured at 700 nm and was directly proportional to its reducing power. Figure 4D shows that the reducing powers of CA-g-CS and CGA-g-CS were significantly (*p* < 0.05) higher than that of CS at the same concentration, indicated by increases in the absorbance by 94.8% and 94.16%, respectively.

### 3.8. Antimicrobial Activity of Phenolic Acid-g-CSs

The antibacterial properties of CS and its derivatives are mainly due to the electrostatic interaction between several positively charged amino groups on the polymer and negatively charged phospholipids on the cell surface, which destroys the cell membrane and leads to cell death [52,53]. According to Figure 5A, the antibacterial activity of the derivatives was superior to that of CS. The MICs of CA-g-CS against *E. coli*, *S. aureus*, and *B. subtilis* were 4, 4, and 4 mg/mL, while those of CGA-g-CS against *E. coli*, *S. aureus*, and *B. subtilis* were 2, 1, and 2 mg/mL, respectively. The antibacterial effect of CS on *S. aureus* was better than the effects on *E. coli* and *B. subtilis*. After grafting with phenolic acid, its antibacterial effect was improved to varying degrees, possibly due to the antibacterial effect of the phenolic acid itself. The incorporated phenolic acid in the CS combines with teichoic acid or lipopolysaccharide on the cell wall, which changes the cell membrane structure, affects the cell permeability, and results in the efflux of the cellular contents [54]. The inhibition zone of the derivatives was positively correlated with the concentration (Figure 5B). Compared with CS, CA-g-CS and CGA-g-CS had more obvious inhibitory effects on the three bacteria. At a low concentration (10 mg/mL), the inhibition effects of CA-g-CS and CGA-g-CS on *E. coli* were not significantly different. At 20 mg/mL, the inhibition zones of CGA-g-CS against *E. coli* (19.16 ± 0.35 mm), *S. aureus* (16.33 ± 0.91 mm), and *B. subtilis* (16.24 ± 0.05 mm) were larger than those of CA-g-CS (16.91 ± 0.44 mm, 11.63 ± 0.34 mm, 14.01 ± 0.4 mm), indicating that CGA had better antibacterial effects than CA. The application of CS is limited due to its relatively weak antibacterial activity. An effective method of enhancing the antibacterial effect of CS is by the chemical modification of its molecular groups. As mentioned above, the grafting ratio affects the antibacterial activity of the conjugate, and they are positively correlated, so that the higher the grafting ratio is, the stronger the antibacterial ability of the phenolic acid-CS copolymer is. Therefore, CGA-g-CS has good antibacterial properties.

In summary, the antioxidant and antibacterial activities of CA-g-CS were lower than those of CGA-g-CS, which may limit its practical application. Therefore, CGA-g-CS was selected to treat the Saimaiti apricot.

### 3.9. In Vivo Assay

Weight loss is an important index used to evaluate the efficacy of the storage and preservation of fruits. It is related to respiration and water evaporation through the epidermal pores [55]. The weight loss of the apricots increased during storage (Figure 6A). Compared with the CK, the CS and CGA-g-CS treatments significantly ameliorated the weight loss in the apricots. However, CGA-g-CS showed a better effect (slower weight loss) than CS at the same concentration. On day 35, the lowest weight loss was exhibited by the CGA-g-CS samples (at 0.5%) at 12.49%, which was 4.32% lower than that caused by CK. These results indicated that CGA-g-CS might act as a barrier on the fruit surface and inhibit the water loss caused by transpiration and respiration [56].

With the passage of the storage time, the firmness of the apricots in all the treatment groups decreased (Figure 6B). This is possibly due to the reduced intercellular adhesion and cell wall mechanical strength caused by cell-wall-degrading enzymes, such as β-galactosidase, polygalactosidase, and pectin methyl esterase [57]. During the whole storage period, the firmness of the fruits treated with CS and CGA-g-CS was higher than that of those treated with CK, while the firmness of the 0.5%-CGA-g-CS-treated fruits was consistently significantly higher than that of the CK-treated fruits (*p* < 0.05). This result was consistent with the previously reported study on the inhibition of the fruit softening process by the application of a chitosan coating [58]. On day 35, the fruit decline rates in the 0.1% CS, 0.5% CS, 0.1% CGA-g-CS, and 0.5% CGA-g-CS treatment groups were 52.17%, 44.62%, 49.24%, and 41.61%, respectively. They were higher than those of CK, at 7.6%, 24.57%, 14.18%, and 31.35%, respectively (*p* < 0.05). These results indicated that all four of the treatments delayed the decline in the fruit firmness. However, 0.5% CGA-g-CS was more efficient in delaying the decrease in the fruit firmness than the CS-only treatment.

When the storage period was extended, the TA content in CK decreased significantly (*p* < 0.05) by 40% on day 7 (Figure 6C). After day 7, the rate of the decrease slowed down, and the content reached 38.45% of that observed at the end of the storage period (day 35). The change in the trend of the TA content in the fruits from the different treatment groups with the storage time was similar to that of CK. On day 7, the TA content decreased by 31.16–38.27%, and at the end of the storage period, the TA content of the treatment groups was 39.4–46.4% of that of the fresh samples. The 0.5% CGA-g-CS significantly delayed the decline rate of the TA in the apricots (*p* < 0.05), and the TA content was 20.67% higher than that of CK by day 35. This is consistent with a previous report [58]. The TA content is reduced mainly because the organic acids are used as major substrates for respiration and other metabolic processes during storage [59].

During storage, the SSC content of the apricots in all the treatment groups increased at first and then decreased (Figure 6D). This trend was mainly due to the conversion of starch and other macromolecular carbohydrates into sugars, organic acids, and other soluble substances during the metabolic process [60]. The SSC content of the apricots in the CK group decreased to 14.44% ± 0.37% by day 35. Compared with the treatment of 0.1% CS, 0.5% CS, 0.1% CGA-g-CS, and 0.5% CGA-g-CS, the results were decreased by 13.12%, 12.7%, 17.15%, and 15.46%, respectively (*p* < 0.05). This is because of the continual respiration, resulting in the decomposition of soluble sugars into CO_2_ and H_2_O [61].

Relative electrical conductivity is the main index of the membrane permeability, and it also indicates fruit ripening and softening [62] (Figure 6E). When the storage time was increased, the relative electrical conductivity of the different treatment groups increased. Compared with the CK, the relative electrical conductivity of the apricots treated with 0.1% and 0.5% CGA-g-CS decreased by 30.05% and 36.4%, respectively, while that of the apricots treated with 0.1% and 0.5% CS decreased by 19.9% and 27.39%, respectively. These results indicated that CS and CGA-g-CS significantly inhibited the softening of the fresh apricots by effectively maintaining the cell membrane permeability.

The respiration rate of the fruits in each treatment group increased first and then decreased during the storage period (Figure 6F). On day 14, the CK-treated fruits displayed a respiratory peak, which decreased slowly, showing the typical fruit characteristics of respiratory jump. The respiratory peaks of the CS and CGA-g-CS treatment groups were delayed by 14 days compared with that of the CK group. It was reported that the respiration rate of coated fruit is lower than that of controls, which may be due to the change in the internal atmosphere (CO_2_ and O_2_) of the fruit, without anaerobic damage [63]. The peak values of 0.5% CGA-g-CS were 15.74%, 14.1%, and 15% lower than those of 0.1% CS, 0.5% CS, and 0.1% CGA-g-CS, respectively (*p* < 0.05). This is because CS and CGA-g-CS formed a film on the apricot’s surface, which inhibited its respiration, delayed the respiratory jump time, reduced the respiration intensity, and delayed the senescence process. This improved the storage and preservation of the fruits [64].

## 4. Conclusions

In this study, we prepared phenolic acid-g-CSs using the H_2_O_2_/Vc radical grafting method. The UV-Vis absorption spectra showed that the grafted copolymer had an obvious ultraviolet absorption peak in the range of 210–500 nm. The FT-IR spectra showed that the CS and phenolic acid were covalently linked by amide and ester bonds. The X-ray diffraction showed that the graft significantly reduced the semi-crystalline properties of CS, and nuclear magnetic resonance spectroscopy and thermal stability analysis further verified that the phenolic acid was successfully grafted onto the framework of CS. Through the antioxidant and antibacterial activities of phenolic acid-CS, it was found that the biological activity of CS was enhanced after phenolic acid grafting. The grafting ratios of CA-g-CS and CGA-g-CS were 126.21 mg CAE/g and 148.94 mg CGAE/g, respectively. The antioxidation and antibacterial activities were positively correlated with the grafting ratio; thus, CGA-g-CS had a stronger inhibition effect on the growth of *E. coli*, *S. aureus*, and *B. subtilis* than CA-g-CS. The in vivo results showed that both the CS and CGA-g-CS (0.1 and 0.5%) treatments could preserve the apricot at 1 °C, but 0.5% CGA-g-CS could better maintain the firmness, weight loss, SSC, TA, relative conductivity, and respiration rate of the apricot. Our research results show that CGA-g-CS has the potential to be used as a postharvest fresh-keeping agent for fruits and vegetables, which provides a reference for the application of phenolic acids in food packaging. This is only a preliminary verification of the effect of phenolic acid-chitosan on apricot preservation. A study aiming to improve the grafting ratio of phenolic acid-chitosan and identify the optimal concentration for its fruit application will be our next research work.

## Figures and Tables

**Figure 1 foods-11-03548-f001:**
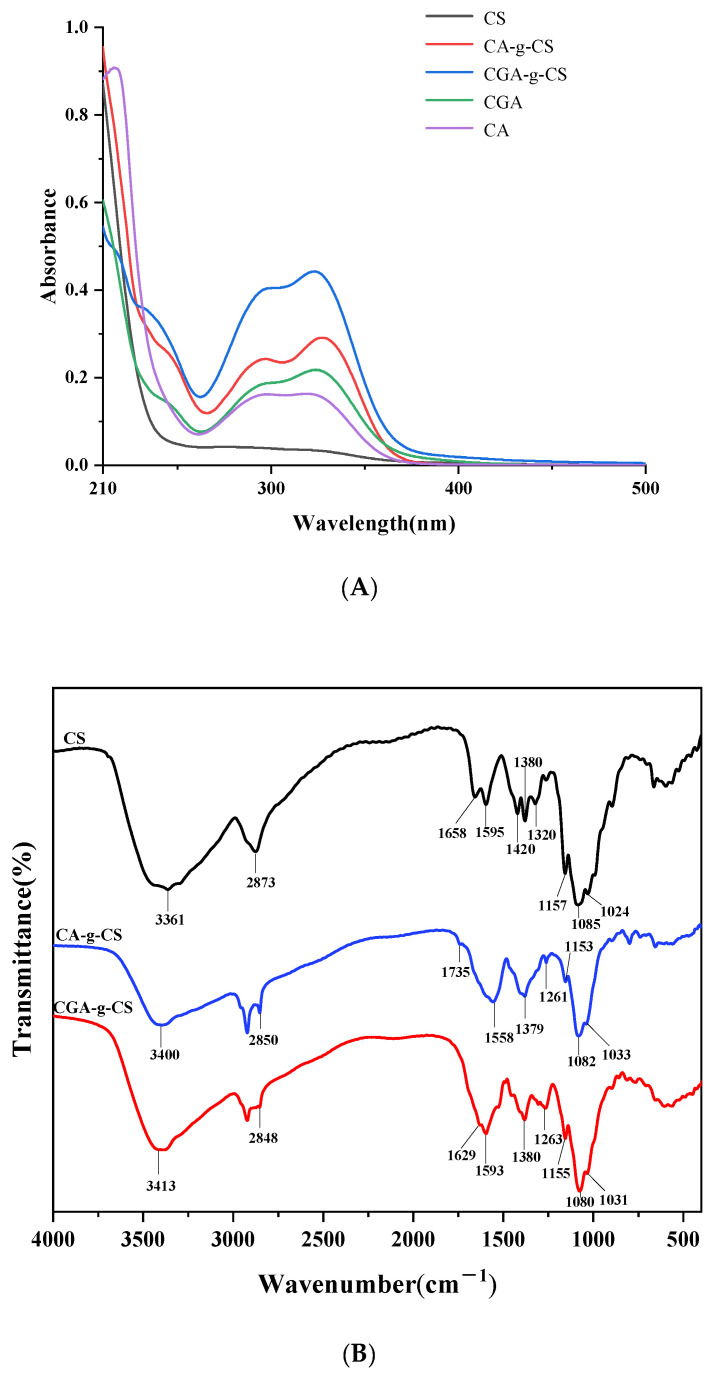
UV–Vis spectra of CS and phenolic acid-CS (**A**); FT-IR spectra of CS and phenolic acid-CS (**B**).

**Figure 2 foods-11-03548-f002:**
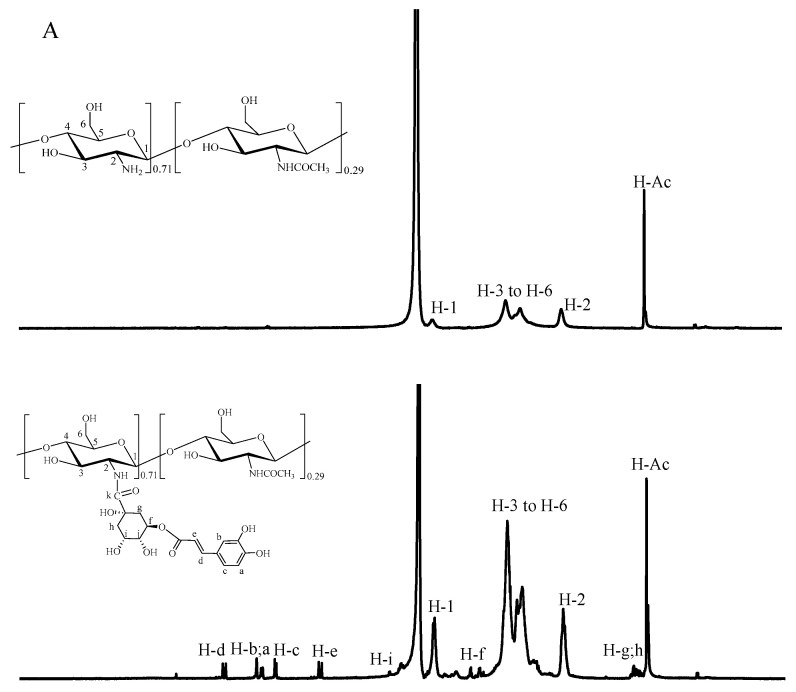
^1^H NMR spectra of CS and phenolic acid-CS (**A**); ^13^C NMR spectra of CS and phenolic acid-CS (**B**).

**Figure 3 foods-11-03548-f003:**
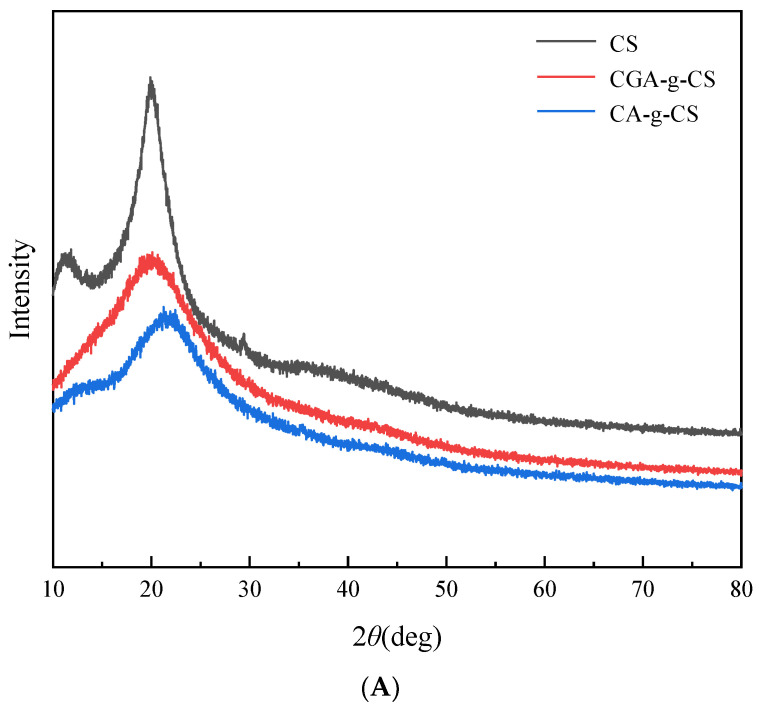
XRD spectra of CS and phenolic acid-CS (**A**); TGA curves of CS and phenolic acid-CS (**B**).

**Figure 4 foods-11-03548-f004:**
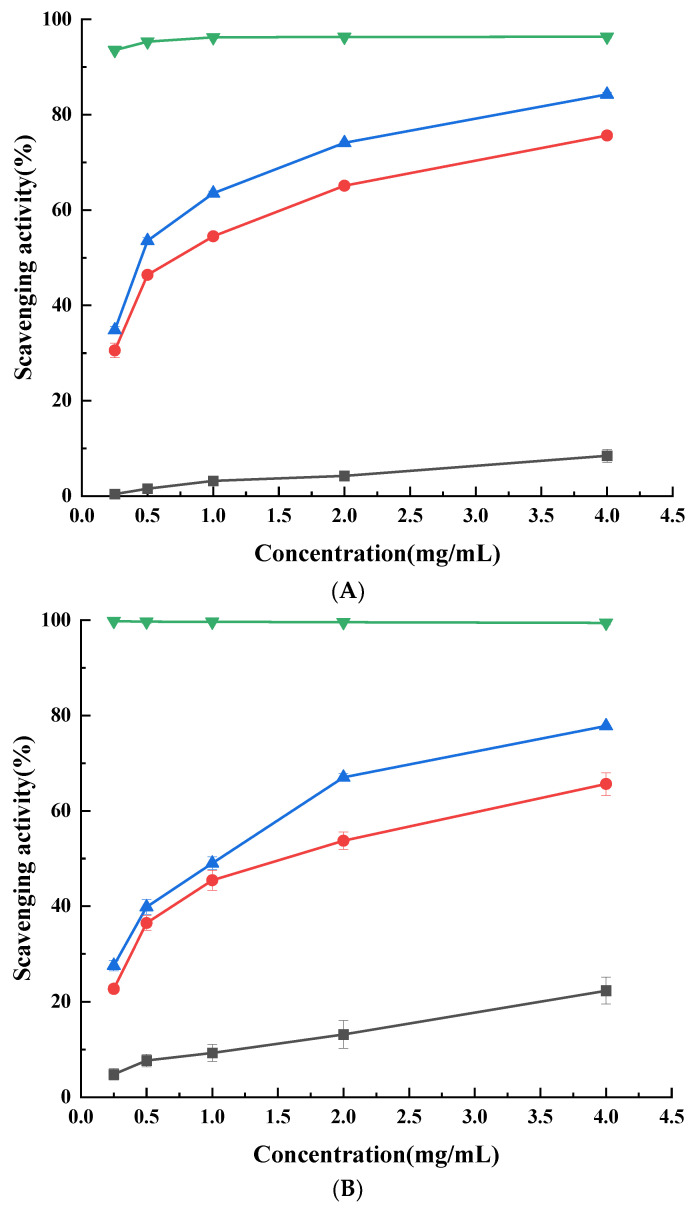
The DPPH radical (**A**), ABTS radical (**B**), hydroxyl radical (**C**), and reducing power (**D**) of CS (■), CA-g-CS (●), CGA-g-CS (▲), and Vc (▼). Data are presented as means ± SD of triplicates.

**Figure 5 foods-11-03548-f005:**
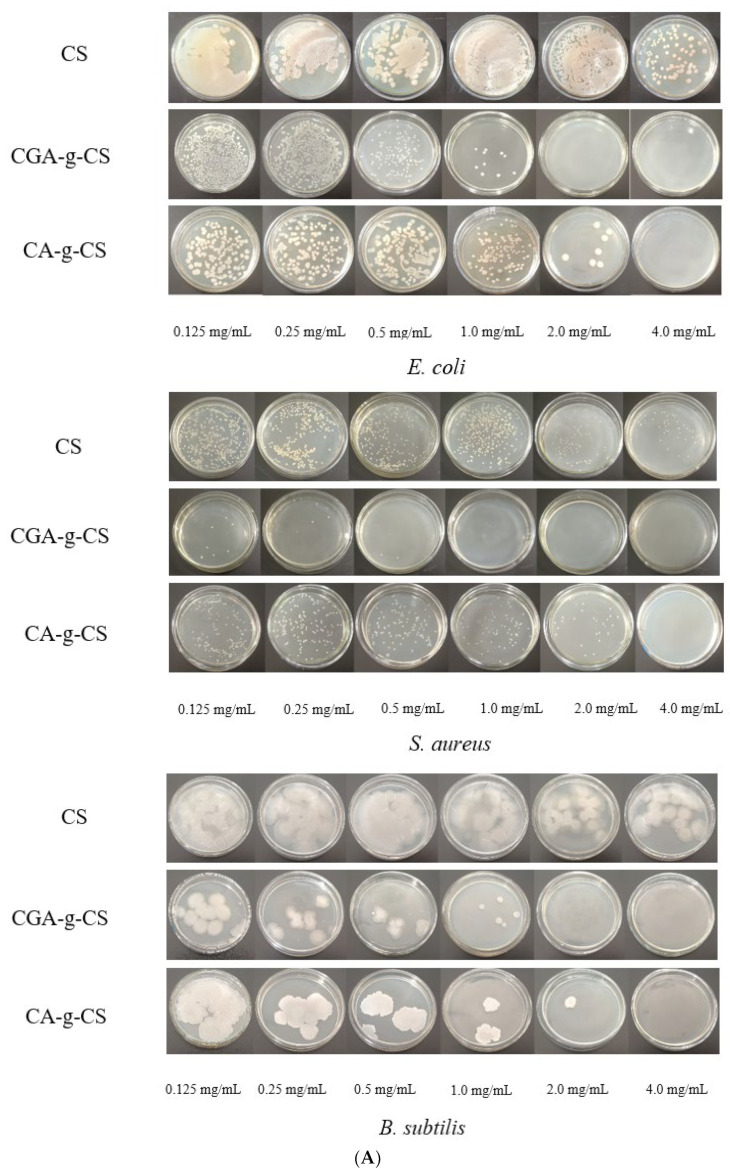
Minimum inhibitory concentration (MIC) of CS and phenolic acid-CS (**A**); inhibitory zone of CS and phenolic acid-CS (**B**). Different lowercase letters in the same column indicate significant differences (*p* < 0.05).

**Figure 6 foods-11-03548-f006:**
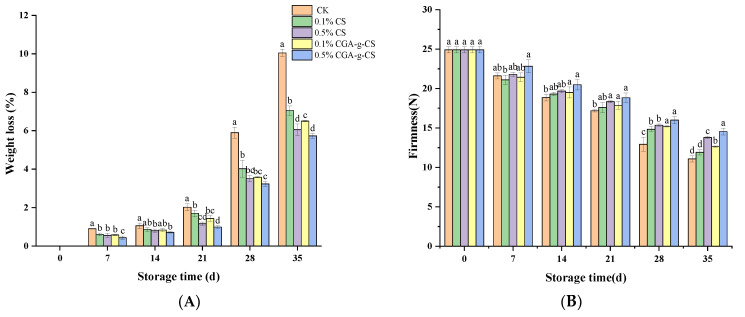
Effects of coatings of chitosan (CS) and CS grafted with chlorogenic acid (CGA-g-CS) on the weight loss (**A**), firmness (**B**), TA (**C**), SSC (**D**), relative electrical conductivity (**E**), and respiration rate (**F**) of apricots during storage at 1 °C for 35 days. The vertical bar represents the standard deviation of the mean (*n* = 3). Values followed by different letters indicate significant differences according to Duncan’s multiple range test (*p* < 0.05).

## Data Availability

All the data of the study can be obtained in this manuscript.

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
