# Peer review of "Preparation and Characterization of Phenolic Acid-Chitosan Derivatives as an Edible Coating for Enhanced Preservation of Saimaiti Apricots"

_foods, 2022, doi:10.3390/foods11223548_

Round 1

Reviewer 1 Report

The paper is well written but significant improvement must be performed. Discussion is very poor and inadequate, so it shoud be drasticaly changed. More issues reffer to zhe following:
Lines 86-87 microorganism should be written in italics and with clearly stated collection (ATCC for example) and reference number. In opposite the source of isolation and identification method must be presented.
Subtitle Results should be changed to Results and discussion

Lines 142-133 only the abbreviated names of microorganisms should be written.

Antimicrobial activity should be investigated against the moulds since they are a more common cause of the deterioration of fruit. Please include these results?

In section 3.7 the EC50 value should be determined and statistical analysis should be included in the paper.
Line 319-320 Please explain your statement in a more detailed way.
MIC for B. Subtilis was also 4 mg/ml.
The discussion of 3.7. and 3.8. should be more comprehensive.
How can you explain lower weight loss for CGA-CS since all the chitosan coatings create a barrier on the surface? Additionally, there is no significant difference between CS and CGA-CS. Why CA-CS was not investigated on apricots?
The conclusion is too ambitious and not supported by the results. The difference between CS and CGA-CS is not always statistically significant so, in my opinion, it can not clearly justify the use of CGA-CS. Please explain your statement.

Reviewer 2 Report

Dear Editors and authors, 

1- The manuscript abstract needs to add some results because the summary is a detailed explanation of the manuscript, and here the abstract is unclear and does not reflect the results of the manuscript. It must be rewritten.

2-The introduction in the manuscript needs to add some scientific references, especially in the paragraphs related to chitosan see line 40-44, and I suggest adding (Al-Manhel, A. J., Al-Hilphy, A. R. S., & Niamah, A. K. (2018). Extraction of chitosan, characterisation and its use for water purification. Journal of the Saudi Society of Agricultural Sciences17(2), 186-190.)

3-Scientific names of bacteria should be written in italics throughout the manuscript, see page 2 line 86 and page 3 line 133.

4- Some methods such as Structural characterization of CA-g-CS and CGA-g-CS needs to add the reference , I suggest (Preethi, S., Abarna, K., Nithyasri, M., Kishore, P., Deepika, K., Ranjithkumar, R., ... & Bharathi, D. (2020). Synthesis and characterization of chitosan/zinc oxide nanocomposite for antibacterial activity onto cotton fabrics and dye degradation applications. International Journal of Biological Macromolecules164, 2779-2787.‏)

5-Antimicrobial activity assay is unclear , This method needs a detailed explanation, how was the bacteria activated? How many bacteria in the inoculation used? What is the volume of the  inoculation added in the culture media?  What is the culture media used?

6-In vivo assay needs to add new reference.

7-There is no statistical analysis in the manuscript and the method used in analyzing the data obtained from the results is not mentioned.

8-Delete Figure 1 from the manuscript. What kind of way did this form come from? This is not a results.

9-The author did not mention the range used for the UV-vis spectrophotometer, see page 3 line 113.

10-The author mentions the range used in FTIR from 4000 - 400, but in the results we find the range from 4000 - 500, see figure 2B.

11-The criteria used in in vivo experiments are not important and the authors had to use other criteria such as the toxicological effect of the compound used.

12-The conclusions contain many consequences and must be rewritten again.

Round 2

Reviewer 1 Report

Thank you for your answers. 

Author Response

Thank you very much for your comments which make our manuscript more perfect. Academic editors let us have a minor revision, and the minor revision version has been uploaded.

Reviewer 2 Report

Dear Editors, 

The authors have modified all requirements for the manuscript. I consent to the manuscript for publication in its current form. 

Author Response

(The authors gave the same response as above.)
